# Mind-the-gap Part II: Improving quantitative estimates of cloud and rain water path in oceanic warm rain using spaceborne radars

Alessandro Battaglia[1,2,3], Pavlos Kollias[4,5], Ranvir Dhillon[1], Katia Lamer[6], Marat Khairoutdinov[4], and Daniel Watters[1,2]

[1]Department of Physics and Astronomy, University of Leicester, Leicester, UK
[2]National Centre for Earth Observation, Leicester, UK
[3]Politecnico of Turin, Turin, Italy
[4]Stony Brook University, NY, USA
[5]University of Cologne, Cologne, Germany
[6]Brookhaven National Laboratory, Upton, NY, USA

**Correspondence:** Alessandro Battaglia
ab474@le.ac.uk

**Abstract.** The intrinsic small spatial scales and low reflectivity structure of oceanic warm precipitating clouds suggest that millimeter spaceborne radars are best suited to provide quantitative estimates of cloud and rain liquid water path (LWP). This assertion is based on their smaller horizontal footprint, high sensitivities, and a wide dynamic range of path integrated attenuations associated to warm rain cells across the millimeter wavelength spectrum, with diverse spectral responses to rain and cloud partitioning.

State-of-the-art single-frequency radar profiling algorithms of warm rain seem to be inadequate because of their dependence on uncertain assumptions on the rain/cloud partitioning and of the rain microphysics. Here, high resolution cloud resolving model simulations for the Rain in Cumulus over the Ocean field study and a spaceborne forward radar simulator are exploited to assess the potential of existing and future spaceborne radar system for quantitative warm rain microphysical retrievals. Specifically, the detrimental effects of non-uniform beam filling on path integrated attenuation (PIA) estimates, the added value of brightness temperatures ($T_B$s) derived adopting radiometric radar modes, and the performances of multi-frequency PIA and/or $T_B$ combinations when retrieving liquid water path partitioning into cloud (c-LWP) and rain (r-LWP) are assessed. Results show that 1) Ka and W-band $T_B$s add useful constraints and are effective at lower LWPs than the same frequency PIAs; 2) matched-beam combined $T_B$s and PIAs from single/multi-frequency radars can significantly narrow down uncertainties in retrieved cloud and rain liquid water paths; 3) the configuration including PIAs, $T_B$s and near surface reflectivities for the Ka-W band pairs in our synthetic retrieval can achieve rmse better than 30% for c-LWPs and r-LWPs exceeding 100 g/m$^2$.

## 1 Introduction

Warm rain is precipitation that originates from non ice-phase processes usually in clouds whose tops lie below the atmospheric freezing level. Warm rain is the dominant mechanism for precipitation formation over the tropical oceans outside the Inter Tropical Convergence Zone, accounting for slightly more than 30% of the total rain amount and 70% of the total rain area in

the 30° S-30° N belt (Lau and Wu, 2003). It has been shown that warm precipitation significantly contributes to moistening and heating of the lower troposphere through latent heat release over most of the tropical and subtropical oceans, thus partially counteracting the low-level cooling induced by stratiform rain (Kodama et al., 2009). In addition, it can impact the organization of shallow convection (Savic-Jovcic and Stevens, 2008; Wang and Feingold, 2009; Yamaguchi and Feingold, 2015; Zhou et al.,

2017). There is also evidence that warm rain impacts cloud lifetime (Paluch and Lenschow, 1991) and even causes changes in cloud regimes; Stevens et al. (1998) noted that, in the presence of strong precipitation, shallow stratocumulus clouds tend to dissipate. Since these direct and indirect mechanisms ultimately affect the global radiative budget and the hydrological cycle (Takahashi et al., 2017; Testik and Barros, 2007), it is important to accurately monitor and quantify the oceanic warm rain variability and improve its representation in large scale models (Stephens, 2005; Dufresne and Bony, 2008).

Global observations of warm rain (Lau and Wu, 2003; Liu and Zipser, 2009; Berg et al., 2010; Kodama et al., 2009) rely on space-borne radar observations (Battaglia et al., 2020): the W-band (94 GHz) Cloud profiling radar (CPR) operated from CloudSat (Tanelli et al., 2008) and the Ku-Ka dual frequency (13.8 and 35.5 GHz) precipitation radar radar (DPR) on board the Global Precipitation Measuring (GPM) Mission core observatory (Skofronick-Jackson et al., 2017) and, before 2014, on the Ku-band PR operated on Tropical Rainfall Measurement Mission (TRMM) (Kummerow et al., 1998). A number

of studies (e.g.Schumacher and Houze (2003); Berg et al. (2010); Rapp et al. (2013); Lamer et al. (2020)) have suggested that none of these spaceborne sensors can describe the entire spectrum of precipitation since, depending on their operational wavelength, such radars are sensitive to precipitation either light (e.g. CloudSat-CPR) or moderate and heavy (e.g. TRMM-PR) and depending on their pulse length they may be of limited use near the surface ($< \approx$500-1000 m height) due to surface clutter and low vertical resolution ($\geq$250 m).

NASA's CloudSat's CPR offers excellent sensitivity (better than -27 dBZ) combined with the smallest footprint ( 1.4 km) amongst existing spaceborne radar missions. Thus, the CPR measurements are used as a reference for the occurrence of light precipitation and drizzle over the oceans (Stephens et al., 2010; Ellis et al., 2009) assuming that a sensitivity of -15 to -25 dBZ in required to detect drizzle (Kollias et al., 2011).

In the part I of our studies (Lamer et al., 2020), focused on spaceborne radar ability to detect and locate presence and

boundaries of low-level clouds and precipitation, determined that surface clutter limits the CloudSat-CPR's ability to observe true cloud base in $\sim 52\%$ of the cloudy columns it detects and true virga base in $\sim 80\%$, meaning the CloudSat-CPR often provides an incomplete view of even the clouds that it does detect. Vertical profiling of such cloud systems down to the closest few hundred meters to the surface with vertical resolution of 250 m or less has been deemed feasible with the next generation of radars, given that they operate with an interleaved shorter pulse (Kollias et al., 2007).

The adoption of aforementioned sampling strategies could lead to a substantial improvement in our ability to map the vertical structure of warm rain. However, quantitative retrievals using single frequency radar measurements remain challenging (Lebsock and Su, 2014; Battaglia et al., 2020). Retrievals of the profile of liquid water content (Lebsock and L'Ecuyer, 2011; Leinonen et al., 2016; Awaka et al., 2016) mainly rely on radar reflectivity profiles. Since reflectivities are sensitive only to the largest and most reflective particles they provide information only on the rain component. When single-frequency mea-

surements are used, the retrieval of rain profiles is inherently affected by large uncertainties related to the a-priori assumptions

on the rain microphysics and its vertical variability. For instance, in the majority of retrievals, a constant-with-height intercept parameter and an exponential drop size distribution (DSD) are assumed. The prospect of using multi-wavelength radar measurements can provide additional information about the vertical structure of cloud and precipitation and thus relax such assumptions (Battaglia et al., 2019, 2020). Using CloudSat observations Rapp et al. (2013); Wang et al. (2017) suggest that the onset of precipitation occurs for cloud liquid water path exceeding values ranging between 150 to above 300 g/m$^2$; c-LWP is then expected to grow with r-LWP (see Fig. 6 in Lebsock et al. (2011)). The cloud affects the radar signal only by attenuating the measured reflectivities: since attenuation amounts to circa 0.8 to 4 and 12.0 dB per kg/m$^2$ of column-integrated cloud liquid content when moving from 35 to 94 and 220 GHz only at the highest frequencies there is a tangible effect. Furthermore, the challenge to detect the cloud base height (Lamer et al., 2020) limits our ability to constrain the vertical extend of the cloud liquid.

Additional constraints to the cloud component come from auxiliary measurements, that may include visible optical thicknesses during daytime and/or path integrated attenuations (PIAs) over the ocean and/or co-located microwave brightness temperatures. Cloud water dominates the visible optical depth with the rain mode only contributing up to 5–10% depending on the details of the DSD (Lebsock and L'Ecuyer, 2011). PIA techniques rely on the contrast between observed surface radar reflectivity under cloud free conditions and under rain, and associate this difference to the attenuation produced by the total water mass in the column. Brightness temperatures ($T_B$) corresponding to oceanic warm rain cells are warmer than the clear sky background due to the additional emission of cloud and precipitation (Wentz and Spencer, 1998; Lebsock and Suzuki, 2016). Lebsock and Suzuki (2016) quantified the typical sensitivities of 94 GHz PIAs and $T_B$ on the total water path to 0.08 K/g m$^2$ and 0.008 dB/g m$^2$ when small values of water paths are considered, with decreasing sensitivities with increasing water paths. They argued that since in the case of CloudSat the PIA precision of 0.16 dB is much better than the 4 K precision of $T_B$ then $PIA$s can potentially achieve a much better retrieval uncertainty, of the order of 20 g/m$^2$ (compared to 50 $g/m^2$) at small water paths.

All the aforementioned integral constrains can substantially improve the estimates of c-LWP and r-LWP; however, they also have drawbacks. Visible optical thicknesses have collocation issues and may be affected by 3D-effects; PIA estimates and $T_B$s measurements are often too noisy at small liquid water paths (LWPs) and strongly affected by non uniform beam filling (NUBF, Kozu and Iguchi (1999); Battaglia et al. (2020); Wentz and Spencer (1998); Lebsock and Suzuki (2016)) associated to the considerable spatial inhomogeneity of warm rain (Lamer et al., 2019).

The relative contribution of cloud and rain particles to the total attenuation depends on the frequency, with the cloud component becoming increasingly important both in absolute magnitude and in relative sense to the rain component with increasing frequency (e.g. see Fig. 2 of Battaglia et al. (2014)). As the radar frequency increases, the size of the small raindrops that contributes the most to attenuation decreases (e.g. the most efficient ones are those of 1.2 mm, 0.48 mm and 0.2 mm radius at 35, 94 and 220 GHz, respectively). These frequency-dependent scattering properties underpin any remote-sensing-based technique for the separation of the cloud and rain component. Besides being essential in retrieval techniques the partitioning between rain and cloud remains an important frontier in improving understanding of the auto-conversion process.

This paper focuses on the quantification of warm oceanic cloud and rain water content using space-borne radars. It aims to:

- discuss the reliability of existing measurements and algorithms and, in the process, identify existing gaps (Sect. 2);

- assess the potential of the future generation of radar observing systems to bridge these gaps, specifically determining (Sect. 3)

  - the impact of different footprints onto non-uniform beam filling effects;

  - if integral constraints for cloud and rain water paths can be improved from using 1) radiometric radar modes; 2) different radar frequencies; and 3) combinations of both types of measurements. These quantities are key in constraining profiling algorithms and their accurate and precise retrieval is *conditio sine qua non* for any quantitative warm rain characterization.

Conclusions are provided in Sect. 4.

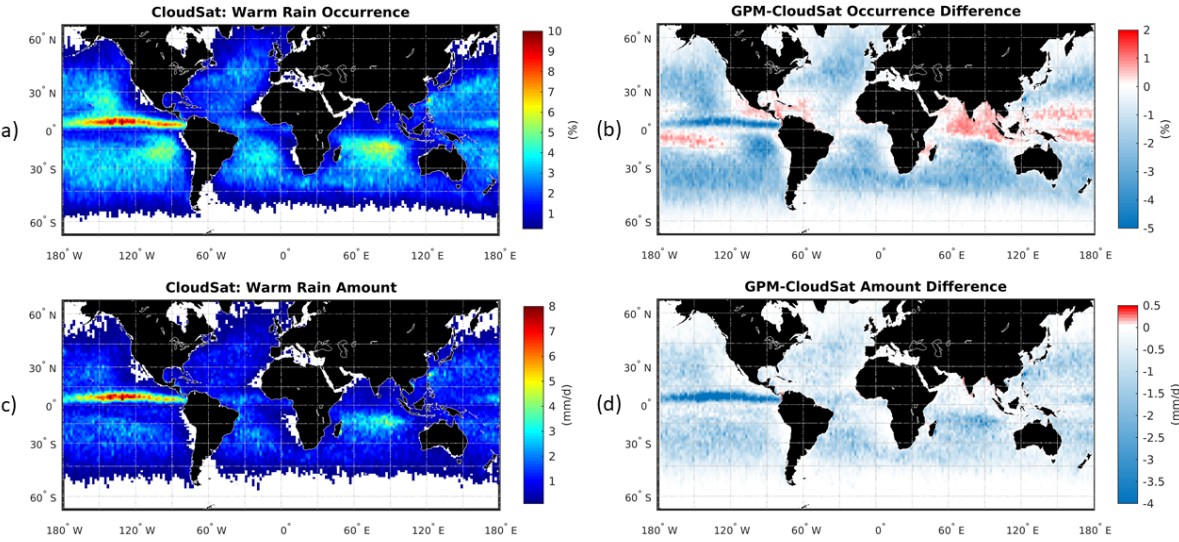

**Figure 1.** Occurrence and amount of global warm rain at the ground as determined by CloudSat (data from January 2007 to December 2010) and GPM (data from July 2014 to June 2018), using a spatial resolution of $2° × 2°$. The CloudSat (GPM) results were produced using the 2C-PRECIP-COLUMN and 2C-RAIN-PROFILE (2A-DPR) products. Panel a) and c): global occurrence and amount of warm rain as observed by CloudSat. Panel b and d): difference in occurrence and amount between GPM and CloudSat.

## 2  Existing gaps

Quantitative estimates of warm rain are currently produced from CloudSat and GPM observations. For CloudSat, CPR reflectivity profiles and auxiliary data from the 2B-GEOPROF are first used to produce a cloud classification in order to identify pixels of warm rain, then exploited to provide profiles of liquid water contents (including the separation between cloud and

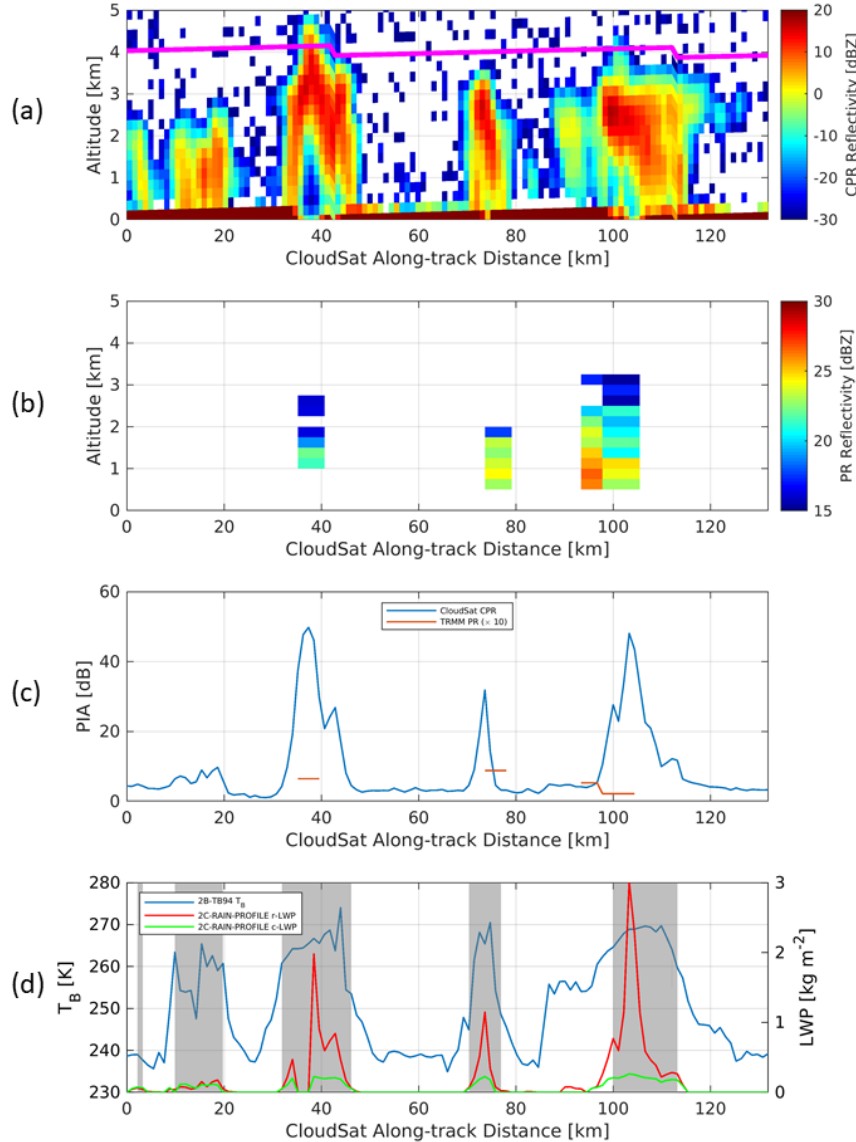

**Figure 2.** Coincident CloudSat and TRMM satellite overpasses over oceanic warm rain cells (2 January 2008, 07:46:46-07:47:05 UT). Panel a): vertical profile of CloudSat radar reflectivities with the magenta line indicating the freezing level height. Panel b): TRMM PR reflectivities. Panel c): CloudSat and TRMM (multiplied by 10) two-way $PIA_{SRT}$s. Panel d): CloudSat brightness temperature. The shaded regions correspond to profiles of warm rain as identified by the 2C-PRECIP-COLUMN.

rain). The 2C-PRECIP-COLUMN algorithm utilizes measurements of the near-surface radar reflectivity and an estimate of the PIA to determine the actual incidence of precipitation. The classification of liquid precipitation into convective, stratiform, or shallow types is then performed by the 2C-PRECIP-COLUMN algorithm (Haynes et al., 2009; Smalley et al., 2014) based

on the vertical structure of reflectivity. In the absence of a reflectivity value of 0 dBZ or greater above the freezing level, the precipitation is classified as shallow.

Two sophisticated algorithms are relevant for warm rain: the 2C-RAIN-PROFILE (Lebsock and L'Ecuyer, 2011) and the 2B-CWC-RVOD product (Leinonen et al., 2016). The algorithms produce rain rates as low as 0.001 mm/h. In contrast to 2C-
RAIN-PROFILE, which is based upon CloudSat radar reflectivities and PIA, the 2B-CWC-RVOD product also includes optical depths derived from the Moderate Resolution Imaging Spectroradiometer (MODIS) on the Aqua A-train satellite (L'Ecuyer and Jiang, 2010).

For GPM, the level-2A Ku, Ka, and dual-frequency precipitation radar (DPR) products provide a warm rain flag (flagShallowRain) and estimates of rain-rate profiles (Awaka et al., 2016). Warm rain is identified within a vertical profile where the
storm top height is more than 1 km below the freezing level (Iguchi et al., 2010). Furthermore, the warm rain is classified as isolated or non-isolated depending upon the horizontal size of the system. In this study, these two categories have been grouped together. Due to the large differences in sensitivity (-29 dBZ vs +12 dBZ for the CloudSat CPR and the GPM DPR respectively), it is reasonable to expect differences in their respective climatology of oceanic warm rain (Liu and Zipser, 2009). Another important factor to consider is the difference in sampling with the CPR providing only nadir pointing observations
while the DPR provides measurements of a 250 km swath.

A CloudSat-based global climatology of occurrence and amount of oceanic warm rain constructed using observations collected between January 2007 and December 2010 is shown in the left panels of Fig. 1. The analysis is restricted to ocean where warm rain is mainly present and because $PIA$ estimates over ocean are much better constrained thanks to the well-predictable ocean surface back-scattering signal (Haynes et al., 2009). A number of features are apparent from Fig. 1. Warm rain is widely
distributed over the Tropics and the Subtropics where the freezing level is located between 4 and 5 km. The occurrence and amount of warm rain are most prominent in the intertropical convergence zone in the East Pacific, where congestus clouds and shallow cumulus clouds below trade inversions coexist with deep convective systems (Johnson et al., 1999). The oceanic warm rain climatology exhibits zonal variability with higher occurrence at warmer sea surface temperatures (SST) that coincide with ascending branches of the Walker circulation. On the contrary, lower occurrences are found near the eastern continental bound-
aries where colder SST's and large scale subsidence result to higher lower tropospheric stability that limits the vertical extend of oceanic clouds and their ability to produce rain.

A similar climatology is derived using GPM DPR observations over the period July 2014 to June 2018. The differences between the GPM and the CloudSat climatology of warm rain occurrence and amount are shown in the right panels of Fig. 1. In general, GPM underestimates both occurrences and amounts. In order to gain insights on the factors contributing to the
observed differences in the ocean warm rain climatology, an example of coincident observations from the CloudSat-CPR and the TRMM-PR is shown in Fig. 2. Note that the same features can be found also when comparing the CloudSat-CPR with the GPM-DPR, since Ku-band TRMM and GPM radars have similar performances in terms of sensitivities and resolution. A number of warm precipitating cloud systems are clearly identified by the CloudSat -PR (reflectivity in Fig. 2a) thanks to its excellent sensitivity and finer resolution. Note that the freezing level is located at about 4 km altitude (magenta line in
Fig. 2a). The Ku-band TRMM radar (see reflectivity in Fig. 2b) has a much coarser horizontal resolution and lower sensitivity

and can detect only the three most heavily precipitating cells of the scene where the CloudSat-CPR is affected by strong attenuation (two-way W-band PIA well exceeding 20 dB, Fig. 2c). The differences in sensitivity and horizontal resolution between current W-band and Ku- and Ka-band systems imply that TRMM/GPM-like spaceborne radars generally struggle in identifying occurrence of drizzle and light warm rain - let alone estimating its intensity- which are well detected by the
CloudSat-CPR. Conversely, high-intensity warm rain, associated with strong attenuation of the CloudSat-CPR signal, is likely to be better quantified by TRMM/GPM-like radars.

There are few tropical regions where GPM overestimates warm rain. This can be either associated with an inconsistency between the two classification algorithms [e.g. GPM warm rain classification may misclassify shallow non-isolated pixels that tend to occur in regions associated with deep convective systems (Funk et al., 2013)] or related to the different resolutions of
the GPM and CloudSat radars. The majority of GPM warm rain cells (55% of the occurrences corresponding to more than 22% of the warm-rain covered area) corresponds to a single GPM pixel. The single GPM pixel resolution is 25 km$^2$ and that of CloudSat 1.5 km$^2$. Thus, a horizontally narrow warm rain with sufficient radar reflectivity to be detected by both sensors can potentially occupy an area 16 times larger in the GPM climatology that it will in the CloudSat climatology. A simple 1D rescaling of the occurrences from the CloudSat 1.1 km along-track resolution to a GPM-like 5.5 km scale accounting for the
reduced GPM sensitivity (with a fixed detection threshold assumed to be 0.02 mm/h, the threshold for warm rain from the 5 km GPM-DPR product) shows that this effect (Fig. 3) explains part of the GPM overestimation in the occurrence statistics. Despite not properly accounting for the 2D averaging, it is clear that the footprint of the instrument (and the minimum detection level) has a large impact on the determination of warm rain occurrences.

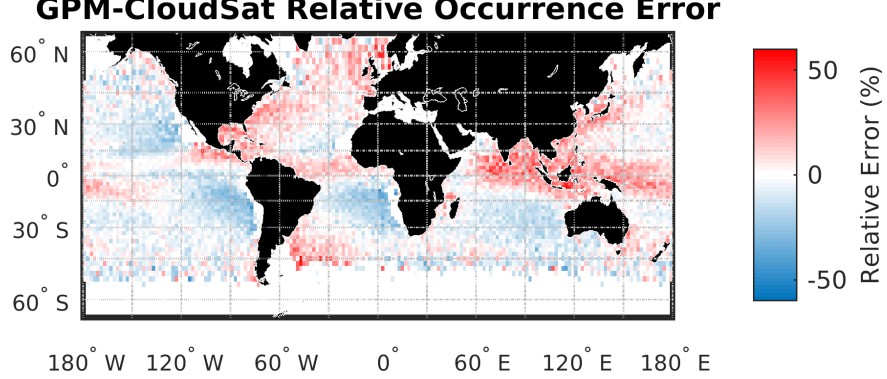

**Figure 3.** Simulated effect onto warm rain occurrences produced by averaging the 1.1 km along-track CloudSat rain rates to a 5.5 km scale and using a 0.02 mm/h detection threshold on the 5.5-km-averaged rainfall. This figure shows the enhancement (positive values) or reduction (negative values) in percentage compared to the CloudSat occurrences.

In addition to the aforementioned challenges related to sensitivity and footprint, recent studies (Christensen et al., 2013;
Mace et al., 2016) concluded that it is very challenging to properly constrain either the cloud or precipitation properties using only single-frequency radar measurements. To further support this, density histograms of cloud liquid water path (c-LWP) and rain liquid water path (r-LWP) derived from the four years (from 2007 to 2010) 2C-RAIN-PROFILE (x-axis) and the 2B-

CWC-RVOD (y-axis) CloudSat products are shown in Fig. 4. The relationship between the cloud and rain water path estimates in these products is very weak. The use of different a priori assumption and the use of different input measurements to retrieve these parameters can explain the observed differences. However, one is left wondering what is the true distribution of c-LWP, r-LWP and their relationship.

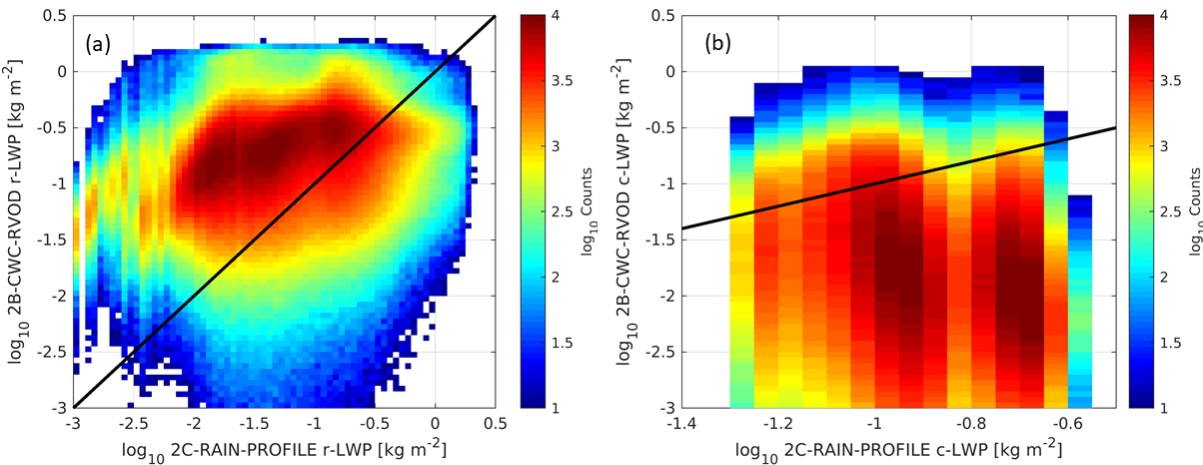

**Figure 4.** Comparisons of (a) precipitation LWP (r-LWP) and (b) cloud LWP (c-LWP) during warm rain (from 2007 to 2010, daytime only) for two CloudSat retrieval algorithms, 2B-CWC-RVOD and 2C-RAIN-PROFILE. Black lines correspond to the 1:1 line: evidence of correlation between the values obtained from the two CloudSat products is minimal.

## 3    The way forward: future space-borne radar configurations

When considering precipitating warm clouds it is clear that future observing systems need to outperform A-Train like observations in order to better constrain precipitation processes. In addition, the scientific community is currently undertaking several notional studies for assessing the potential of different multi-frequency radar concepts (Battaglia et al., 2020), with the core effort at NASA in preparation of the Aerosol-Clouds-Convection-Precipitation (ACCP) mission, following the recommendation of the National Research Council Decadal Survey for Earth Sciences (The Decadal Survey, 2017). Advances in radar technology enable new research avenues, with three aspects of particular relevance for warm rain studies.

– The next generation of space borne radars should achieve *better sensitivities and finer vertical and horizontal resolutions* (see Tab. 1 for two specifications currently under consideration). This work focuses at evaluating the impact of the reduction in horizontal resolution onto quantitative estimates.

– *Multi-frequency radars* including channels within the G-band (Battaglia et al., 2014; Cooper et al., 2018; Roy et al., 2018; Battaglia and Kollias, 2019) are now plausible candidates for constellation concepts. Clearly different frequencies are tailored to different targets, with higher (lower) frequencies more suitable for observing clouds (precipitation) because of their better sensitivity (reduced attenuation), as already demonstrated in Fig. 2.

**Table 1.** Summary of specifications for two radar configurations currently under study: one with a very large antenna antenna and one with a smaller antenna. A flying altitude of 400 km and a range resolution of 500 m are assumed.

| Frequency [GHz] | Configuration I ($4.5 \times 4$ m$^2$) | | Configuration II ($2.5 \times 1.6$ m$^2$) | |
| --- | --- | --- | --- | --- |
| | Single pulse MDS [dBZ] | Footprint size [km$^2$] | Single pulse MDS [dBZ] | Footprint size [km$^2$] |
| 13.6 | 10 | $2.4 \times 2.7$ | - | - |
| 35.5 | -6 | $0.9 \times 1.0$ | 2 | $1.65 \times 2.6$ |
| 94.4 | -23 | $0.36 \times 0.41$ | -17 | $0.65 \times 0.95$ |
| 220 | -25 | $0.14 \times 0.16$ | -19 | $0.28 \times 0.41$ |

– *Radars with radiometric modes* can provide perfectly antenna-matched $T_B$, with the possibility of fully exploiting the combination of active and passive measurements. An example is provided by the CloudSat brightness temperature product (2B-TB94 available at http://www.cloudsat.cira.colostate.edu/data-products, see details in Mace et al. (2016)) derived from processing the noise floor data contained in the 1B-CPR data product. For the case study shown in Fig. 2 the Cloud-Sat $T_B$ (bottom panel) is well correlated with the presence of the cells, with an enhancement of $T_B$s, compared to the cold oceanic background at $\sim$240 K, induced by cloud and rain emission like observed in Advanced Microwave Scanning Radiometer (AMSR) channels at similar frequency (Eastman et al., 2019). Interestingly, $T_B$s seem to produce a quicker response than the PIA to the presence of the rain cells. This is due to two reasons. First, visually, with the scale of values shown in the y-axis of Panel c) and d) it is much easier to see a change in $T_B$ of the order of 1 K than a change in PIA of 0.1 dB (which both corresponds to a water path of roughly 12.5 g/m$^2$). Second, CloudSat $T_B$s are computed by averaging along track using a 5 pixel boxcar window so they have a coarser resolution than that of the PIA.

## 3.1 Methodology

A forward simulator framework using cloud resolving model simulations as input is used to test the impact of horizontal resolution of the different radar configurations and to assess the potential of a multi-frequency radar system (with or without radiometric mode). Two cloud field conditions are evaluated (one with freezing levels around 4.9 km and the other with freezing level around 3.4 km) and a total of 12 different radar configurations were tested with 3 different footprints for each frequency (as tabulated in Tab. 2) which spans the range of values currently employed or expected in the near-future.

### 3.1.1 RICO simulations

The simulations are based on data collected during the Rain in Cumulus over the Ocean (RICO) field study. RICO was a comprehensive field study of shallow cumulus convection which was located in the winter trade winds of the northwestern Atlantic ocean, just upwind of the Islands of Antigua and Barbuda. An overview of the experiment is provided in Rauber et al. (2007). The focus of RICO was on the statistical character of the cloudy boundary layer, particularly on the characterization of precipitation in shallow cumulus. The cloud-resolving model used in this study is version 6.11.2 of the System for Atmospheric

Modeling (SAM) described by Khairoutdinov and Randall (2003). The double-moment Morrison et al. (2005) microphysics was used with prescribed concentration of cloud droplets of 100 cm$^{-3}$. The horizontal doubly-periodic domain size was 57.6×57.6 km$^2$ with horizontal grid spacing of 100 m, while the vertical grid had 120 levels with the domain top at 4.8 km and constant 40 m grid-spacing.

Two simulations have been performed. The set-up of the first simulation closely follows the setup used in the study by vanZanten et al. (2011). The second simulation was for drier conditions, more characteristic of the midlatitudes rather than subtropics. There is no straightforward way to scale the soundings and forcing profiles to accomplish that, so a very simple procedure was used. The original sea-surface temperature of 299.8 K as well as the initial temperature of the sounding were reduced by 9.8 K. The water vapor sounding and prescribed large-scale vapor tendencies were scaled by a factor that was
obtained to keep the relative humidity at the first model level constant. The other profiles, such as the horizontal wind and subsidence rate were not changed. As the result of the procedure, the freezing level moved from around 4.9 km down to 3.4 km height. Each simulation was run for 2 days with time step of 2 seconds.

    Fig. 5 shows one scene (out of the 65 considered in this study) illustrating the low freezing level simulations with the colors modulated by the total integrated c-LWP and the contour lines showing the r-LWP equal 0.1 and 0.5 kg/m$^2$, (black and gray
lines). Instantaneous field of views (IFOV) of planned and operated space-borne radars are shown for reference on the right hand side.

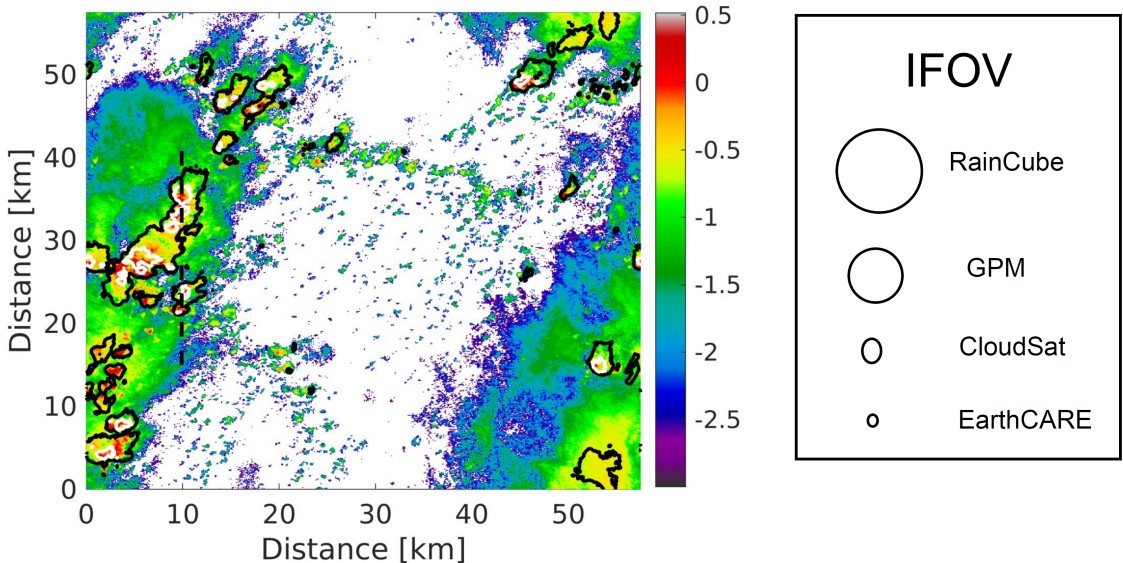

**Figure 5.** Example of a SAM simulation output: cloud liquid water path ($\log_{10}(LWP[kg\,m^{-2}])$) with contour lines of 10 g/m$^2$ and 500 g/m$^2$ rain LWP in black and white, respectively at native model resolution. The black dashed line corresponds to the ground track for the overpass shown in Figs. fig2:RICO-8.

### 3.1.2 Forward radar simulator

Four steps are followed in order to produce radar observables from the model output.

1. Scattering properties of the medium are computed at the fine model resolution. Gas attenuation is computed according to the model from Rosenkranz (1998). Scattering properties of cloud droplet and raindrops are computed via Mie theory (e.g. see Lhermitte (1990)). An exponential drop size distribution is assumed for rain with $N_{0r} = 8 \times 10^6 \ m^{-4}$ (Marshall and Palmer, 1948)whereas a Gamma distribution with $\mu = 3$ with an effective radius increasing from 3 to 15 $\mu m$ from cloud base to cloud top according to Bennartz (2007). The impact of the Marshall and Palmer assumption is discussed later.

2. The scattering properties are used as input of the radar forward model developed in Hogan and Battaglia (2008) for computing the attenuated and unattenuated reflectivity profiles (but with the multiple scattering flag turned off) and of an Eddington approximation code (Kummerow, 1993) for computing $T_B$s.

3. A surface echo is added. The surface normalised backscattering cross section, $\sigma_0$ is computed as a function of the 10 m wind speed and the sea surface temperature following Wu (1990). For the computation of $T_B$s, surface emissivities are computed according to ocean Fresnel models.

4. The results produced in steps 2 and 3 are convolved with the radar range weighting functions with a top hat pulse with 500 m range resolution (see Lamer et al. (2020)) and two-dimensional Gaussian antenna patterns with different instantaneous field-of views. Along track integration of 500 m is carried out.

5. Random noise with zero mean and 2 K (0.7 dB) standard deviation is added to the $T_B$s and the PIAs while reflectivity noise is computed using the dependence on signal-to-noise ratio (SNR) and averaging pulses described in Hogan et al. (2005). Note that, as shown in Leinonen et al. (2017), the uncertainty in the PIA is dominated by the uncertainty of the clear sky $\sigma_0$, not by the uncertainty of the surface return (unless for cases closes to full attenuation) which, at high SNR and for CloudSat configuration, amounts to 0.17 dB (Lebsock and Suzuki, 2016). Leinonen et al. (2017) articulate that the uncertainty in the clear sky $\sigma_0$ can be estimated from the standard deviation of the observed clear-sky surface cross sections, with an average value of 0.24 dB for CloudSat so that the total PIA precision is estimated to be 0.29 dB. In reality this is an optimistic assumption because the clear sky $\sigma_0$ is likely more uncertain, especially if there are no "clear sky" observations in the vicinity of the cloudy/rainy profile, which occurs e.g. when continuous Sc decks are present. In addition, even if such calibration points are available, winds are likely to change in presence of precipitation due to local circulation (e.g. this feature is evident in the SAM simulations, not shown). $\sigma_0$ has a strong sensitivity to the 10-m wind speed (Fig. 6) whereas the sensitivity to SST is ver weak ($< 0.08 dB/K$, not shown). Therefore with 7-10 m/s wind (which are characteristic values in our simulations), an uncertainty of 1 m/s in the wind speed induces an uncertainty of roughly 0.4-0.6 dB in $\sigma_0$ (with much worse results obtained for lower wind speeds). Therefore we generally think that a PIA precision of 0.7 dB is more appropriate to be assumed in our discussion. Similarly nadir emissivity sensitivities

are of the order to 0.2-0.3%/(m/s) at all frequencies here considered. A 1 m/s uncertainties propagates into a maximum 0.6-1 K uncertainties in the $T_B$s.

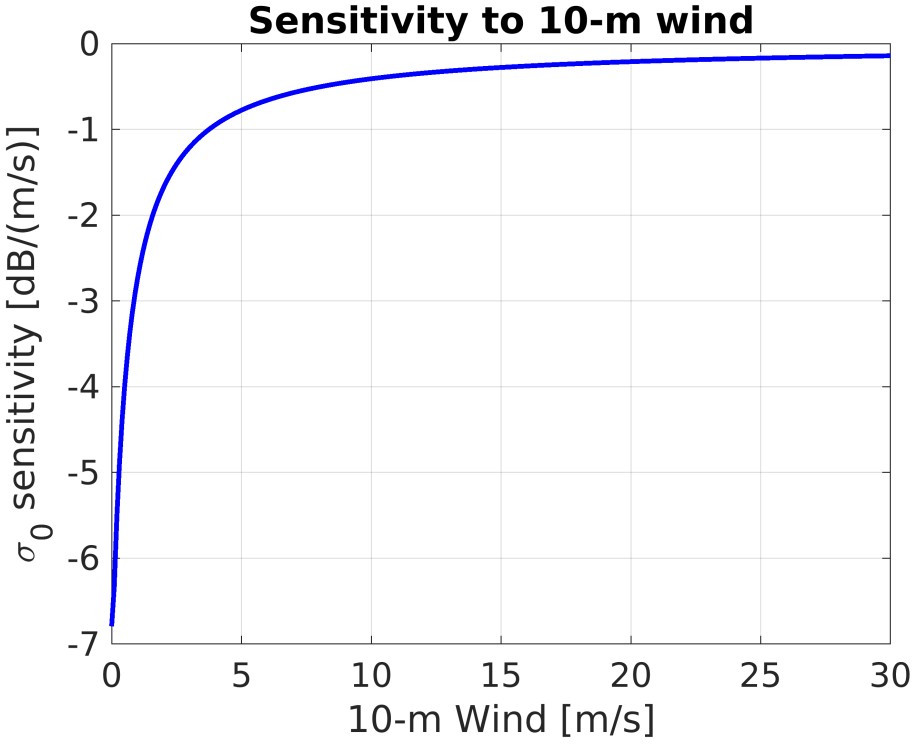

**Figure 6.** Sensitivity of the normalized radar cross section, $\sigma_0$, to 10-m wind speed according to the Wu (1990) model.

6. Since the sensitivity improves proportionally to the antenna gain, the single pulse minimum detection signal (MDS) of each configuration is obtained by scaling the values provided in Tab. 1 for each frequency according to the scaling of the antenna footprint area. Forward simulated reflectivities below the MDS are removed.

### 3.1.3 Example of simulated radar observables

Fig. 7 indicates the integrated cloud and rain LWP, the brightness temperature $T_B$ and the path integrated attenuation PIA along the black line drawn in Fig. 5. The integrated measurements are shown for different horizontal resolutions (model: 0.1 km and radar: 1 and 4 km) and for different radar frequencies (Ku-, Ka-, W- and G-band). The horizontal transect of cloud and rain LWP indicate the presence of two clusters of shallow precipitating clouds with the first one having the majority of its water in cloud size droplets and the latter having the majority of its water in rain size particles. Each cell is approximately 5 km wide and exhibit considerable variability with 2-3 cores spaced by 1.5-2 km. The integrated LWP are shown in the original model resolution and for two different radar footprints (1 and 4 km). Noticeably, the 1-km radar footprint is able to preserve the shallow precipitation spatial variability, while the 4 km footprint results to considerable smearing of the warm rain spatial

distribution. This is consistent with the discussion in Sect. 2. $T_B$ clearly reacts to the emission of the precipitating clouds which warm the cold oceanic background (Eastman et al., 2019). When increasing frequency, the nadir ocean emissivity and the atmospheric gas optical thickness (mainly due to water vapor) rise, thus causing a significant warming of the baseline "clear sky" $T_B$s. This reduces the contrast between clear sky and rainy cells, thus making $T_B$ measurements less useful. On the
other hand, optical thicknesses of the precipitating columns (which are driving the $T_B$ warming) also increase with frequency, thus making the high frequency channels the most sensitive to the presence of rain. For instance, Ku-band $T_B$s have a low baseline and a potentially large dynamic range, but only few K of enhancement are produced in this scene because of the low sensitivity of the Ku-band extinction to cloud and rain (see Battaglia et al. (2020)). Viceversa the W-band baseline $T_B$s are already quite warm due to a high ocean surface emissivity; furthermore they are strongly affected by the amount of water vapor
in the atmosphere, with significantly lower clear sky baselines for environments with lower freezing level (e.g. $\sim$235 K for this and similarly for the scene shown in Fig. 2 compared to $\sim$255 K for the scene shown in the right panel of Fig. 5). $T_B$s at 220 GHz (not shown) are already saturated in clear sky conditions and are therefore not considered further in this paper.

For the same cross section vertical profiles of cloud and rain water content and reflectivities are shown in Fig. 8. The two rain cells located around 23 and 35 km are well profiled by all frequencies, but the configurations with the largest footprints
[Panels b) and d)] tend to fatten their structures. The increased level of attenuation at higher frequencies is clearly highlighted by the reduction in the surface return [Panels e) and f)]; at 220 GHz, the surface return and the 1.5 km lowest rain disappear below the MDS in correspondence to the heavily precipitating core at 35 km.

## 3.2   Results

### 3.2.1   Impact of instrument footprint

Warm precipitating clouds are naturally very inhomogeneous as clearly highlighted in Fig. 5. This inhomogeneity and their inherent 3D structure is very challenging when considering spaceborne observations, with the instrument footprints naturally convolving (thus smoothing) the natural variability of the rain microphysics fields. For instance the sharp peaks in c-LWP and r-LWP at the 100 m model native resolution are already reduced when moving to 1 km footprints and completely washed out when moving to 4 km (top panel in Fig. 7). Similar effects can be seen in $T_B$s and PIAs (e.g. compare lines with the same colors
and different styles in the centre and bottom panels in Fig. 7) and also in the reflectivity profiles with rainy cells appearing less intense but with larger horizontal extent for the coarser horizontal resolution radar configurations (compare left and right centre panel in Fig. 8).

NUBF effects are particularly detrimental when considering estimates of (two-way) PIA via the surface reference technique (hereafter designated PIA$_{SRT}$) and in the implementation of attenuation corrections for measured reflectivities within profiling
algorithms [Durden (2018) and reference therein]. The PIA$_{SRT}$ is derived by contrasting the surface return under rainfall and the surface return under clear-sky conditions (Nakamura, 1991; Meneghini et al., 2015). In equations

$$PIA_{SRT} \equiv 10\log_{10} \frac{\iint_{\Omega} G^2(\hat{\mathbf{\Omega}}) e^{-\frac{pia(\hat{\mathbf{\Omega}})}{4.343}} d\Omega}{\iint_{\Omega} G^2(\hat{\mathbf{\Omega}}) d\Omega} \tag{1}$$

whereas the "true" PIA is obtained via averaging the integrated attenuation along each direction $\mathbf{\Omega}$ within the footprint, $pia(\mathbf{\Omega})$:

$$PIA \equiv \frac{\iint_\Omega G^2(\hat{\mathbf{\Omega}})pia(\hat{\mathbf{\Omega}})\,d\Omega}{\iint_\Omega G^2(\hat{\mathbf{\Omega}})d\Omega} \tag{2}$$

where the integrals are extended to the full antenna pattern and $G$ represent the antenna gain. If $pia(\mathbf{\Omega})$ is constant across the footprint then $PIA_{SRT} = PIA$ but in presence of NUBF $PIA_{SRT}$ always underestimates the true PIA, as a direct conse-
quence of the concavity of the exponential function. This is clearly demonstrated in Fig. 9 for a Ka-band having a footprint of 1 and 4 km (left and right panel, respectively). Note the presence of negative values for PIA$_{SRT}$ (y-axis) in correspondence to small true PIAs due to the injection of noise in their estimates. The radar footprint can impact the PIA estimates in two different ways:

1. It can lead to considerable underestimation of the PIA due to NUBF effects (i.e. the spatial extend of the cloud covers
only a fraction of the effective radar footprint that is the result of the instantaneous field of view and of the along track integration). This effect is amplified for larger radar footprints (see departure from the 1:1 line in Fig. 9).

2. The range of the true PIA value decreases when coarser footprints are adopted due to the convolution with the broader 2-way antenna gain function. As a result of the smaller measured PIA values, the relative error introduced by the measurement error increases. Thus the fraction of useful PIA measurements strongly increases with greater frequencies and
with smaller footprints (see last column in Tab. 2).

Similar results are found at the other frequencies and are summarized in Tab. 2 where the slope coefficient of the fitting line ($m_{SRT}$), the correlation ($\rho_{SRT}$) and the maximum value of the PIA$_{SRT}$s are reported. With the same footprint, the NUBF effect tend to be more and more acute with increasing frequencies. For current configurations like for the CloudSat-CPR or the GPM-Ka, the value of $m_{SRT}$ are significantly lower than one. It is clear that proper corrections must be applied. While for
the GPM-DPR studies are focused at identifying the pixels mostly affected by NUBF by using the combination of normal and high sensitivity scans (with flags introduced in the "Trigger" module, see Mroz et al. (2018)) and at implementing corrections (Seto et al., 2015), to our knowledge, no mitigation is currently planned for warm rain retrievals in the CloudSat algorithms currently in operation (Lebsock and L'Ecuyer, 2011; Leinonen et al., 2016). The same applies to the EarthCARE CPR (which is less affected having a footprint substantially smaller) algorithms (Mason et al., 2017), now in preparation. A possible solution
would be to identify regions highly affected by NUBF by producing $PIA_{SRT}$ estimates at much higher temporal frequency than cloud measurements (which is possible thanks to the typically higher SNR of the surface) and using the along-track PIA gradient as a proxy for NUBF (something similar is currently envisaged for the Doppler NUBF correction, e.g. see Kollias et al. (2014)). The estimated gradient of $PIA_{SRT}$s could be used to partially correct for the bias. Fig. 10 shows the relationship for a 94 GHz radar with a 0.5 km IFOV. There is still quite some spread around the median value (black line), as expected from the
highly non-linear impact of NUBF on $PIA_{SRT}$s (e.g. see example in Mroz et al. (2018)). For larger footprints the spread of the biases around the median tend to increase (not shown). Most of the non-linearities are expected to be particularly acute in presence of footprints with considerable empty fractions. Future work should investigate the role of ancillary visible or infrared

**Table 2.** NUBF effect on PIA estimates based on the surface reference technique. For all footprints an along-track integration length of 500 m is assumed. In the last column the fraction of profiles with PIA exceeding 2 dB is referenced to the number of profiles having c-LWP or r-LWP exceeding 10 g/m$^2$ at 0.5 km resolution.

| Frequency [GHz] | Footprint [km] | $m_{SRT}$ | $\rho_{SRT}$ | Max PIA$_{SRT}$ [dB] | Fraction with PIA$\geq$ 2 dB |
|---|---|---|---|---|---|
| 13.6 | 1 | 0.82 | 0.69 | 6.5 | 0.28 |
| | 2.5 | 0.66 | 0.14 | 4.5 | 0.06 |
| | 4 | 0.49 | 0.01 | 3.2 | 0.02 |
| 36.5 | 1 | 0.62 | 0.85 | 28.1 | 3.7 |
| | 2.5 | 0.49 | 0.63 | 11.7 | 2.2 |
| | 4 | 0.44 | 0.41 | 6.1 | 1.0 |
| 94.0 | 0.5 | 0.66 | 0.90 | 76.1 | 16.3 |
| | 1.0 | 0.55 | 0.84 | 59.8 | 14.7 |
| | 2.5 | 0.41 | 0.67 | 25.2 | 10.3 |
| 220.0 | 0.5 | 0.62 | 0.86 | 62.1 | 33.1 |
| | 1.0 | 0.52 | 0.78 | 59.9 | 30.3 |
| | 2.5 | 0.38 | 0.63 | 36.0 | 22.7 |

collocated images or of scanning/push-broom configurations with oversampled footprints to better quantify the footprint empty fraction.

### 3.2.2 Value of radiometric measurements vs PIAs

In order to better assess the value of $T_B$s a simple scenario is considered, with the inclusion in the environment of the case study of Fig. 5 of a cloud layer at roughly 2 km and a rain layer underneath with increasing c-LWP and r-LWP values. The corresponding brightness temperatures for cloud-only (crosses) and rain-only (lines) conditions are plotted in Fig. 11. Different microphysics with exponential distributed DSD are considered for the rain. In addition to the Marshall and Palmer we have added another configuration with constant intercept $N_0 = 5 \times 10^7$ m$^{-4}$ which is labelled as "Cumulus" since it resembles the one labelled with the same name in Lebsock and L'Ecuyer (2011). In addition the SAM model actually adopts a two-moment scheme where rain concentration and rain water contents are both predicted Morrison et al. (2005); such scheme is approximated by a power law fit of the form $\Lambda^{-l} = 2.7 \times 10^{-4} RWC^{0.3}$ where $\Lambda$ is the slope parameter in m$^{-1}$ and RWC is the water content in kg/m$^3$ (and it will be referred as "MO05"). From these plots the different response of the different frequencies is immediately clear: while the 13.6 GHz never saturates for LWP smaller than 5 kg/m$^2$ the 36.5 and 94 GHz saturates at different c- and r-LWP. The sensitivity of the $T_B$s to a change in cloud and rain LWP is illustrated in the botom panel of Fig. 11. The sensitivity to cloud is generally smaller than to rain, but such difference becomes increasingly smaller with increasing frequencies; similarly for the differences between different rain microphysics. This is a direct consequence of the dependence of the extinction per unit mass on the characteristic size of the DSD (see Fig. A6 in Battaglia et al. (2020)).

The SAM simulation can reach very low cloud and rain LWP (down to $10^{-4.5}$ kg/m$^2$ and $10^{-5.5}$ kg/m$^2$, respectively). An indication about how the two quantities co-vary when considering quantities averaged over footprints of the order of 1 km is provided by the color image in Fig. 12. Fig. 12 further illustrates where PIA and $T_B$ can be useful as constraints for retrievals. In order to gauge the relevance of PIA and $T_B$ measurements, the mean values of $\Delta T_B$s (defined as the enhancement of the $T_B$s with respect to their clear sky reference values) and PIAs in correspondence to different pairs of (r-LWP, c-LWP) binned logarithmically within their range of variability have been computed. A threshold of 0.7 dB for hydrometeor PIAs and of 2 K for $\Delta T_B$s have been used to roughly identify regions where those signals are sensitive to variations of c-LWP and r-LWP. The corresponding contour lines are shown in Fig. 12 for PIAs (dashed) and $\Delta T_B$s (continuous lines) for the different frequencies (different colors). We can draw the following remarks.

– Ka and W-band $\Delta T_B$s and PIAs (see green and blue curves) are potentially very useful; similarly PIA at G-band. PIA Ku-band appears effective only for heavily precipitating clouds (i.e. $r - LWP \geq 400$ g/m$^2$). The inclusion of increasing frequencies increases the retrieval potential towards smaller values of c-LWP but, even considering G-band, integral constraints are useful only for c-LWP and r-LWP exceeding roughly 30 g/m$^2$ and 15 g/m$^2$, respectively.

– For any given frequency, $T_B$s are generally useful for smaller cloud and/or rain contents compared to PIAs (if precision of 2 K and 0.7 dB are assumed for $T_B$s and PIAs). For instance at 35 GHz for non-raining (raining) clouds, $T_B$s are certainly useful for all $c - LWP > 60$ $g/m^2$ ($r - LWP > 20$ $g/m^2$) but PIAs only when $c - LWP > 350$ $g/m^2$ ($r - LWP > 100$ $g/m^2$). W-band (Ka-band) PIAs seem to have the same (slightly better) potential as Ka-band (than Ku-band) $T_B$s.

– $T_B$s tend to saturate at larger LWPs than the corresponding PIAs (see also Fig. 2). The dynamic range of $T_B$s (defined as the difference between saturated and background $T_B$s) is constantly reducing with increasing frequency (because of the increase in the ocean nadir emissivity and the water vapor optical thickness). At the G-band (not shown) it becomes so small that $T_B$s are practically of no use.

### 3.2.3   Potential of multi-frequency retrievals of c-LWP and r-LWP

Here we want to investigate which frequency and which variable combination is more effective for retrieving column-integrated cloud and rain liquid water paths. The retrieval methodology is based on an Ensemble Kalman smoother approach described by Grecu et al. (2018). The large database of RICO simulations is used to produce statistical relationships between observations and the state variables. Here the vector of unknowns is $\mathbf{x} = [c - LWP, r - LWP]$ while the vector of measurements can include both $PIA_{SRT}$s, $T_B$s and/or reflectivities measured at the "closest to the surface" clutter-free bin, thus $\mathbf{y}_{obs} = [T_B \ldots Z_{ns} \ldots PIA \ldots]$, which can contain multiple frequencies and/or multiple observables. Note that near surface reflectivities, when multi-frequency observations are available, have been suggested as variables with good potential to provide constraints on integral quantities (Durden, 2018).

Given a vector of observations $\mathbf{y}_{obs}$, the retrieved vector of unknowns is computed as:

$$\mathbf{x}_{ret} = \langle\{\tilde{\mathbf{x}}\}\rangle + cov\left(\{\tilde{\mathbf{x}}\},\,\{\tilde{\mathbf{y}}\}\right)\left[cov\left(\{\tilde{\mathbf{y}}\},\,\{\tilde{\mathbf{y}}\}\right)\right]^{-1}\left(\mathbf{y}_{obs} - \langle\{\tilde{\mathbf{y}}\}\rangle\right) \tag{3}$$

where $\{\tilde{\mathbf{y}}\} = [\tilde{\mathbf{y}}_1,\,\tilde{\mathbf{y}}_2,\ldots\tilde{\mathbf{y}}_n]$ is a subset of the simulated observations of the training database within a normalised distance, $\delta$ [see Eq. 4], from $\mathbf{y}_{obs}$ less than one and $\{\tilde{\mathbf{x}}\} = [\tilde{\mathbf{x}}_1,\,\tilde{\mathbf{x}}_2,\ldots\tilde{\mathbf{x}}_n]$ are the corresponding state variables. The operator $\langle\ \rangle$ indicates the mean operation across the subset. The normalised distance between the observation vector and the $j-$element of the subset of $\{\tilde{\mathbf{y}}\}$ is defined as:

$$\delta\left[\mathbf{y}_{obs},\tilde{\mathbf{y}}_j\right] = \frac{1}{length(\mathbf{y}_{obs})}\sqrt{\sum_{k=1}^{length(\mathbf{y}_{obs})}\frac{\left(\mathbf{y}_{obs}(k) - \tilde{\mathbf{y}}_j(k)\right)^2}{\sigma_k^2}} \tag{4}$$

where $\sigma_k$ is 2 K for $T_B$s, 1 dB for PIAs and depends on the SNR for $Z^{ns}$ according to formula A9 in Hogan et al. (2005).

Absolute bias and root mean square errors for different combination of observables at different frequenices with matched footprints of 1 km are plotted in Fig. 13 . It is clear that Ka and W-only retrievals perform poorly, but with the W-band (Ka-band) performing better for the cloud (rain) component. Combining single-frequency PIAs and $T_B$s (circles) certainly improves the retrieval but it is the combination of two frequencies that definitely improve simultaneously c-LWPs and r-LWPs, with the Ka-W performing better than the W-G combination (compare cyan and magenta circles). The inclusion of near surface reflectivities also produces a significant improvement both in the performances of the c-LWPs and the r-LWPs (see dashed lines). The inclusion of the G-band in addition to the Ka and W seems to only marginally improve results (compare dashed cyan and black lines). Note that LWPs are expressed in logarithmic units, thus rms differences of 0.1, 0.2 and 0.3 corresponds to fractional errors of +26%/-21%, +58%/-37%, +100%/-50%, respectively. So for instance the Ka-W band pair achieves best retrieval performances with rmse better than 30% for c-LWP and r-LWP exceeding 100 g/m$^2$. Similar conclusions (not shown) can be drawn when 2.5 km footprints are used.

## 4    Conclusions

Quantification of warm rain from space-borne radars remains challenging because of its intrinsic patchy structure and low reflectivity structure which both impose the use of high frequency radars (Ka and above) capable of small footprints and high sensitivity. State-of-the-art single-frequency radar profiling algorithms of warm rain seems yet inadequate because of the dependence on uncertain assumptions on the rain/cloud partitioning and of the rain microphysics. Such assumptions can be mitigated if multi-frequency observations and additional integral constraints (specifically PIAs derived from the surface reference technique and brightness temperatures acquired in radiometric mode) are considered. In this paper, the impact of such measurements on the retrieval of the cloud and rain integrated liquid water paths has been examined. The findings can be summarized as following.

1. PIA constraints are generally extremely useful in retrieval algorithms. However, for RICO-like scenes, characterised by extreme non uniform beam filling conditions, the surface reference technique provides biased estimate of the PIA even

when sub-km footprints are considered. Simulated PIAs for CloudSat- and GPM-like radars are seriously affected. To our knowledge, no attempt is currently made in CloudSat (and EarthCARE) retrievals to correct for such an effect. This could potentially produce negative biases in LWP estimations. The possibility of adopting corrections e.g. driven by VIS and IR imaging should be contemplated. Also scanning configurations that oversample the footprints could be considered in order to effectively improve the spatial resolution by deconvolution (Battaglia et al., 2020); this is particularly true at Ka-band to achieve sub-km like effective footprints.

2. Increasing frequencies enhances the PIA sensitivity towards smaller values of LWP. W-band produces PIA exceeding 0.7 dB with c-LWP (r-LWP) exceeding about 70 g/m$^2$ (20 g/m$^2$). A G-band channel can significantly boost sensitivity because of its stronger cloud extinction coefficient, thus lowering the detection threshold by a factor of circa 2.5 compared to W-band.

3. Over ocean surfaces, $T_B$s are generally more sensitive to smaller cloud and/or rain contents compared to PIAs if precisions of 0.7 dB and of 2 K are assumed for PIAs and $T_B$s. Lebsock and Suzuki (2016) have assumed worse precision in the $T_B$s (4 K) based on the CloudSat product (which is an experimental product, the radar was not designed for having a radiometric mode) and much better precision in PIAs ($\approx 0.16\ dB$), thus pushing the PIA cloud sensitivity limit at W-band to 20 g/m$^2$. In our analysis, given the uncertainties in the retrieval of the ocean surface $\sigma_0$ in rainy conditions, we have used a more conservative approach. The precision limit when retrieving c- and r-LWP via $T_B$s can be directly computed by looking at Fig. 11 for any given $T_B$ measurement precision.

4. $T_B$s steadily increase due to cloud/rain emission but they tend to saturate at LWPs values which are smaller and smaller with rising frequencies. The dynamic range of $T_B$ variability (defined as the difference between saturated and background $T_B$s) is constantly reducing with increasing frequency (because of the increase in the ocean emissivity and in the water vapor optical thickness). At 200 GHz and above, they are of no practical use.

5. If matched beams are accomplished, dual-frequency constraints adopting PIAs and/or $T_B$s significantly improve c-LWP and r-LWP retrievals compared to single frequency. The Ka-W band pair achieves the best retrieval performances with rmse better than 30% for c-LWP and r-LWP exceeding 100 g/m$^2$. Near surface reflectivities further improves accuracy and precision of the retrieval.

Future work should thoroughly assess the retrieval capabilities of the full vertical structure of warm rain from dual and triple frequency (Ka, W and G-band) matched-beam observations. In general, non-Rayleigh differential scattering effects for the Ka-W and W-G combinations are huge for rain (up to 10 and 15 dB, respectively for $D_m$ up to 1 mm, Battaglia et al. (2020)). This gives opportunity to size raindrops if reflectivity profiles remain above the noise level [thus sufficient MDS must be reached] and if attenuation is properly accounted for. This will help in reducing the uncertainties related to the rain microphysics; in such framework a rigorous assessment of the impact of the different rain/water partitioning in various microphysical schemes could also be carried out. Note that G-band reflectivities are generally expected to be useful mainly in the upper part of the warm rain profile (where the cloud is expected to be). At such frequencies, in high freezing level condition (3 km and above), gas and

hydrometeor attenuation and non-Rayleigh effects tend to drive the signal below detection threshold in the lower troposphere. Two aspects remain critical for substantial progress of warm rain profiling capabilities:

– the inclusion of good a-priori assumptions, especially related to the distribution of the cloud liquid water given the break down of adiabatic assumption in rainy conditions (Li et al., 2015; Zhu et al., 2019);

– the incorporation of 3D and NUBF effects. This requires a paradigm change by moving from retrieving each single profiles separately, to running simultaneously retrievals for the entire cross section of the warm rain cell as captured by the profiling radar suite. Retrievals must be adapted to address the intrinsic 3D nature of warm rain systems.

*Code and data availability.*

All GPM and CloudSat data used in this paper are available via NASA portal.

*Author contribution.*

AB and PK designed the study layout. CloudSat data analysis was performed by RD. GPM analysis was performed by DW. SAM simulations were perfomed by MK. AB did the simuation analysis and prepared the manuscript with contribution of all co-authors.

*Competing interests.*

The authors declare that they have no conflict of interest.

*Acknowledgements.* This work was supported by the European Space Agency under the "RainCast" activity, Contract No. 4000125959/18/NL/NA. The work by Alessandro Battaglia has been supported by the project Radiation and Rainfall funded by the UK National Center for Earth Observation (RP18G0002). This research used the ALICE High Performance Computing Facility at the University of Leicester.

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

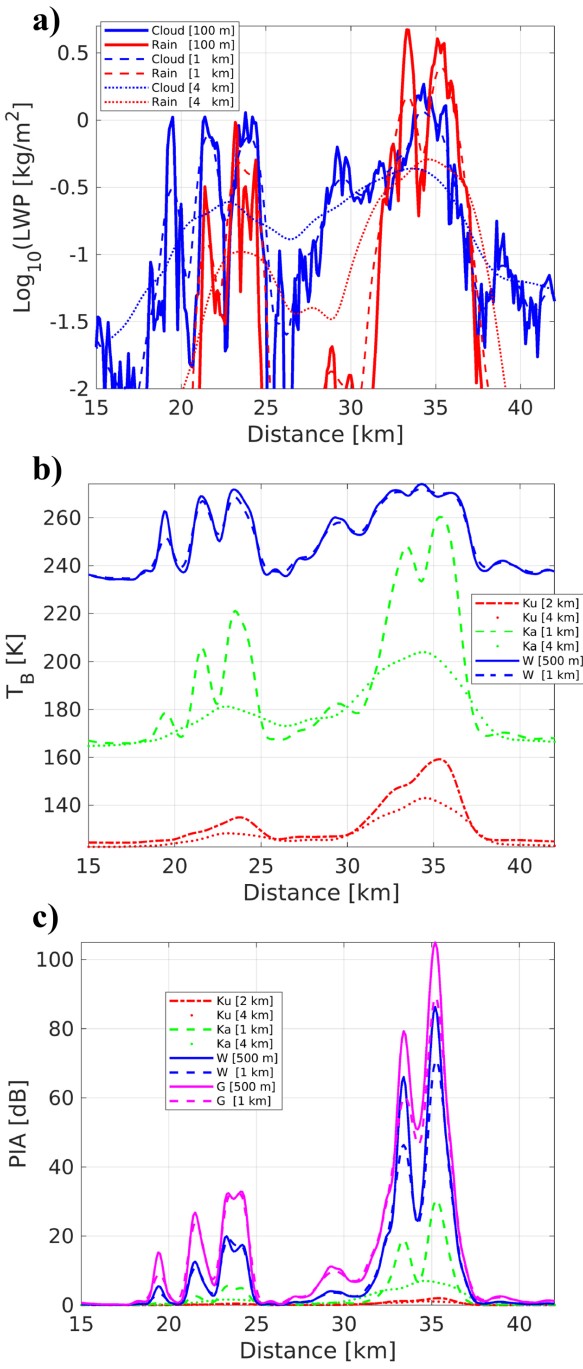

**Figure 7.** Panel a): integrated cloud and liquid water path along the cross section shown in the left panel of Fig. 5 at the native model resolution and when accounting for different footprint sizes, as indicated in the legend. Panel b): brightness temperatures for different frequencies and footprint sizes as indicated in the legend. Random noise with a standard deviation of 2 K is added to the measurements. A nadir looking geometry is assumed. Panel c): two-way hydrometeor PIAs computed simulating SRT observations for different frequencies as indicated in the legend. Red, green, blue and magenta color correspond to Ku, Ka, W and G-bands.

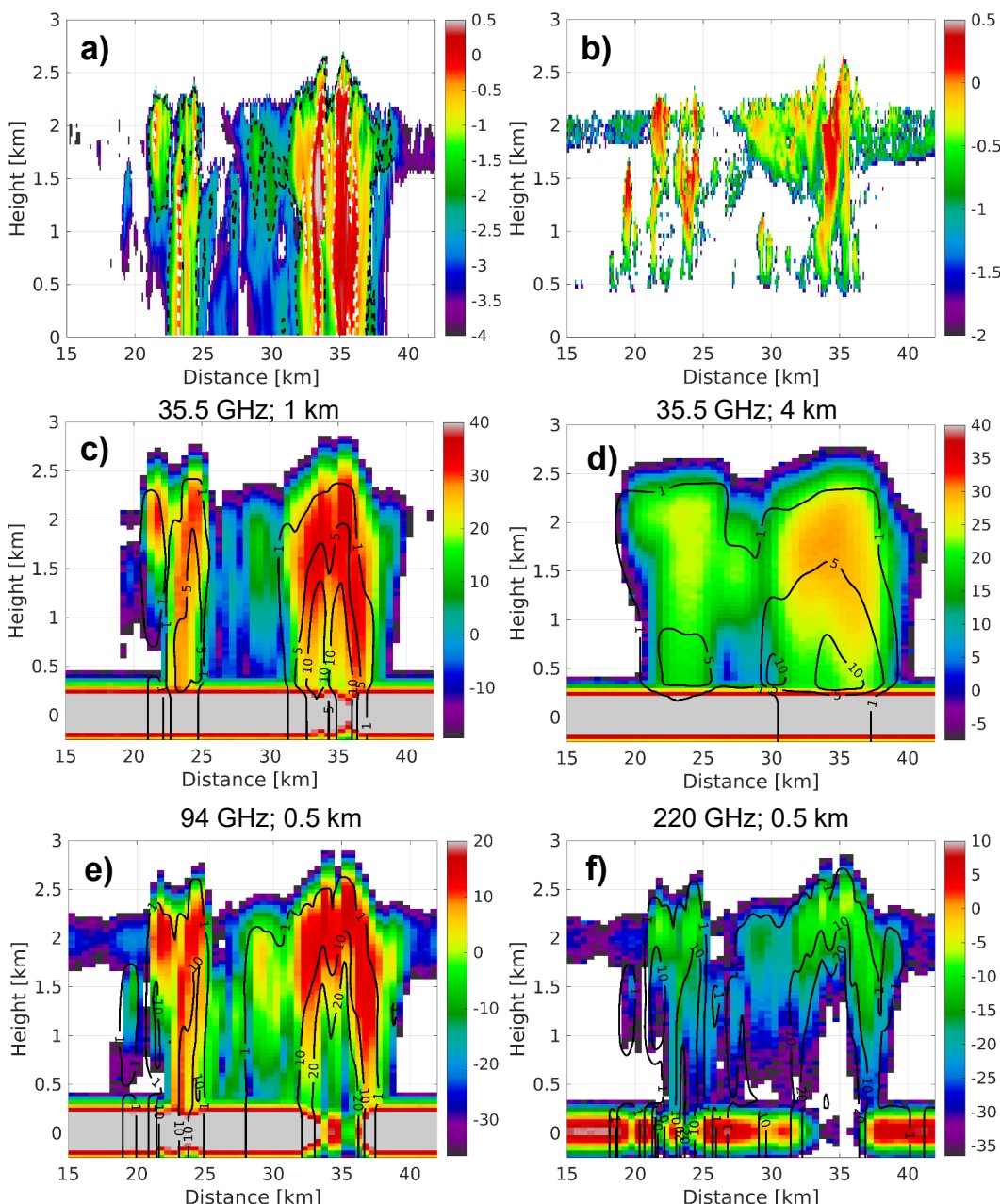

**Figure 8.** Panel a): vertical profile of r-LWC for the cross section corresponding to the black line shown in Fig. 5 at the model resolution (100 m); the black and white dashed lines correspond to $D_m$ equal 0.5 and 1.5 mm, respectively. Panel b): vertical profile of c-LWC for the same cross section. The units of the colorbar for both top panels corresponds to $\log_{10}$ of LWC in g/m$^3$, so for instance -1 means 0.1 g/m$^3$. Panels c-g): simulated vertical profiles of reflectivities (colorbar units in dBZ) for bands and horizontal resolutions as indicated in the text in the top left of each panel. Black lines correspond to contour levels of the antenna weighted PIA from the top of the atmosphere.

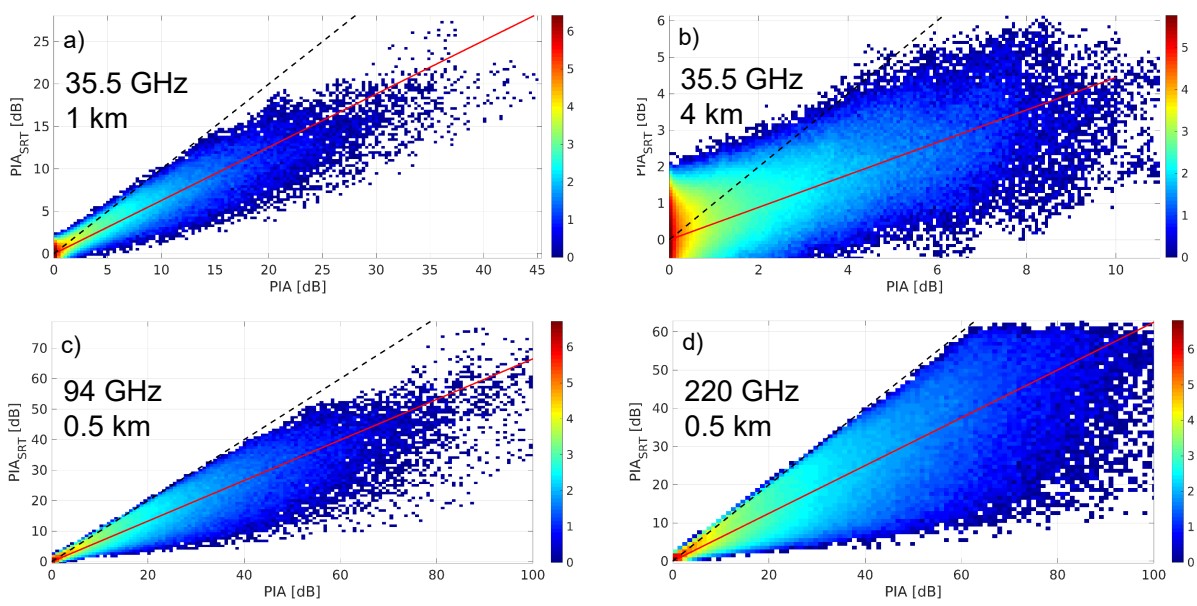

**Figure 9.** Illustrating the impact of NUBF onto estimate of the hydrometeor PIA via the surface reference technique: density scatterplot of hydrometeor PIA vs $PIA_{SRT}$ for the whole dataset of RICO simulations for Ka-band radars with footprint of 1 km (a) and 4 km (b) and for footprints of 0.5 km at 94 GHz (c) and 220 GHz (d).

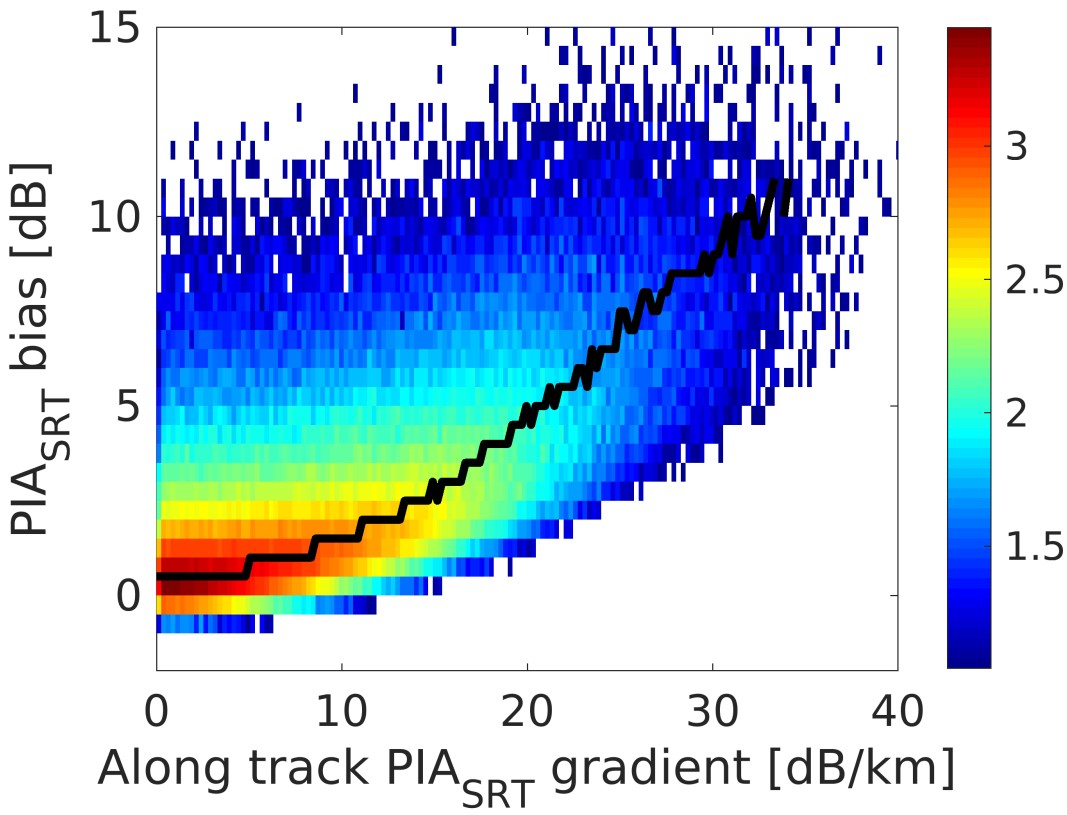

**Figure 10.** Density plot of $PIA_{SRT}$ bias (with respect to the true PIA) as a function of the absolute value of the along-track $PIA_{SRT}$ gradient for the entire dataset of simulations. A configuration for an EarthCARE-like configuration is considered (94 GHz radar with a 0.5 km IFOV). The black line corresponds to the modal values of the biases for a given absolute value of the $PIA_{SRT}$ along-track reflectivity

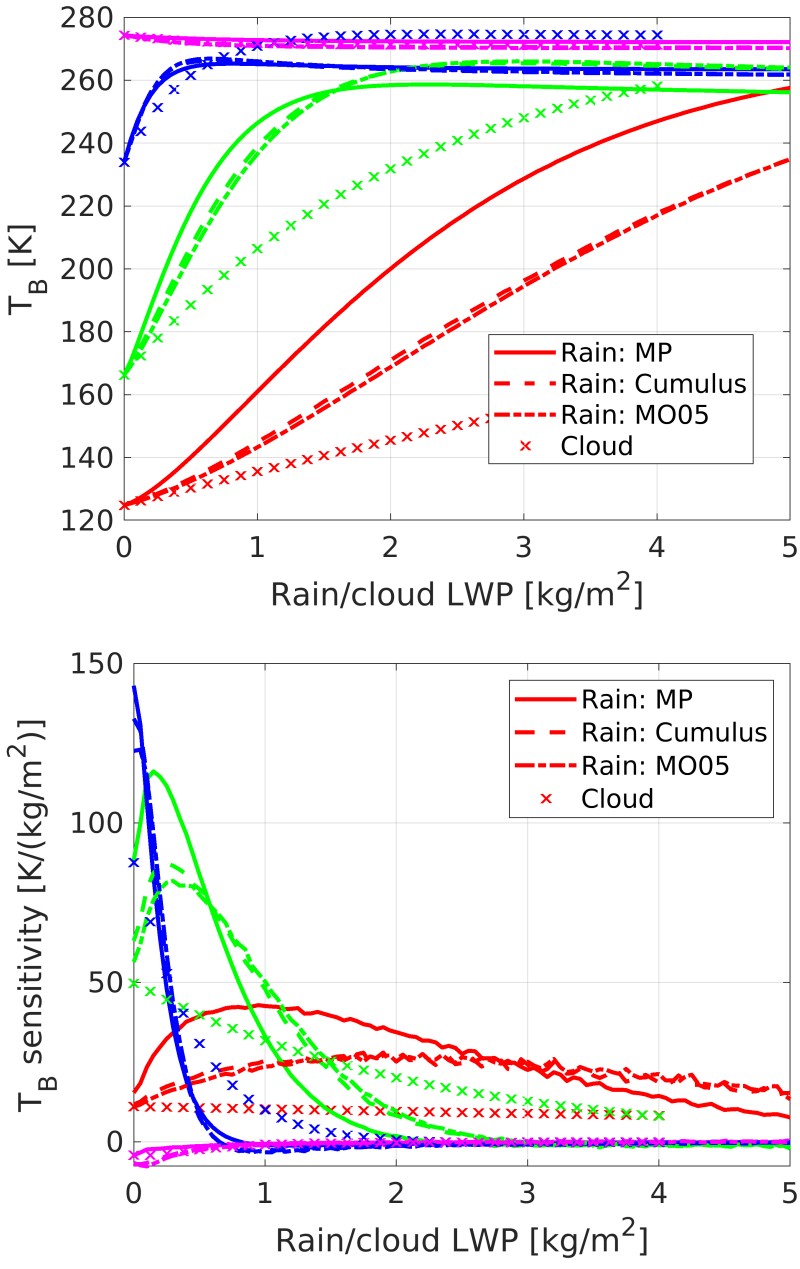

**Figure 11.** $T_B$s (top) and $T_B$-sensitivity (bottom) as a function of the c-LWP and r-LWP for a simple scenario with cloud-only and rain-only conditions. Different microphysics scheme are considered for rain (see text for details). Red, green, blue, magenta lines correspond to 13.6, 36.5, 94 and 220 GHz.

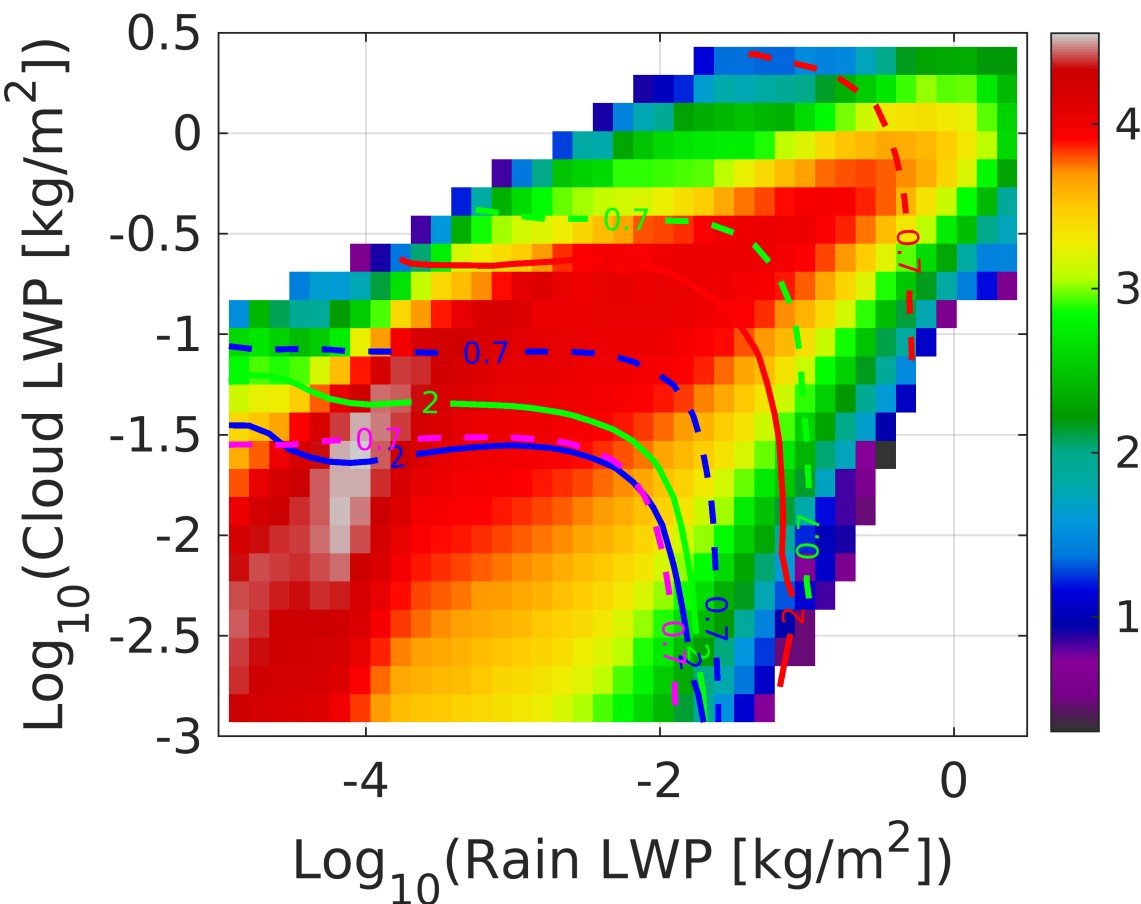

**Figure 12.** Distribution of r-LWP vs c-LWP with $\log_{10}$ of the occurrences modulated by the colorbar. Footprints of 1 km are used for all instruments. The continuous lines (unless otherwise labelled) corresponds to where $T_B$s are exceeding the background $T_B$s by 3 K, whereas dashed lines corresponds to where the hydrometeor PIAs are exceeding 1 dB. Red, green, blue, magenta lines correspond to 13.6, 36.5, 94 and 220 GHz.

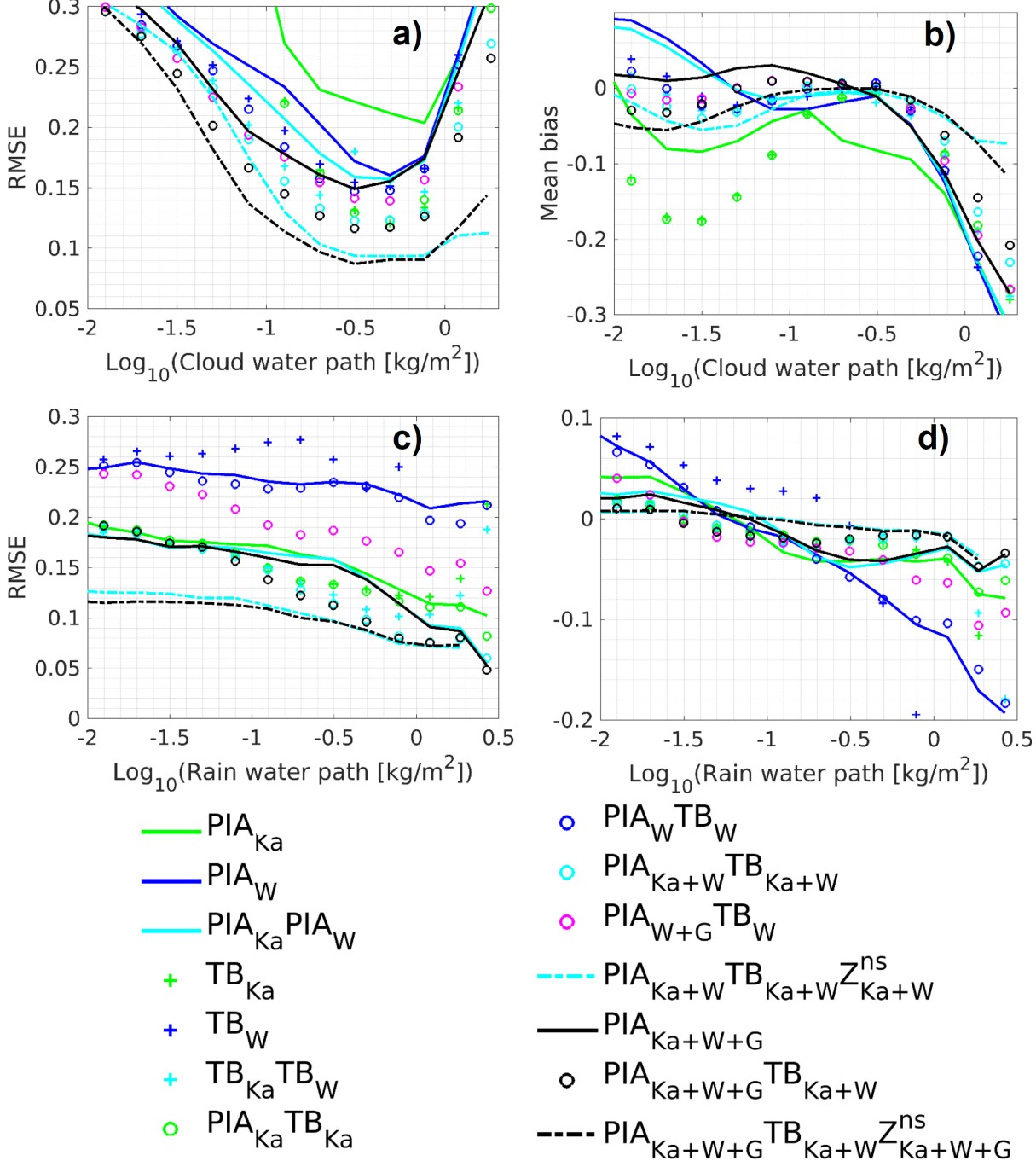

**Figure 13.** Bias (left) and rmse (right) in retrievals of c-LWP (top) and r-LWP (bottom) in $\log_{10}$ units for the configuration with 1 km footprints. A value 0.1, 0.2, 0.3 correspond to a factor 1.26, 1.58 and 2 error. Continuous lines, crosses, circles and dashed lines correspond to PIA-only, $T_B$-only, PIA and $T_B$ combined, PIA, $T_B$ and $Z^{ns}$ combined, respectively. Green, blue, cyan, magenta and black lines correspond to Ka-only, W-only, Ka-W, W-G and Ka-W-G combination, respectively.