# Peer review of "Mind-the-gap Part II: Improving quantitative estimates of cloud and rain water path in oceanic warm rain using spaceborne radars"

_Atmospheric Measurement Techniques, 2020_

## Referee Comment (RC1) · Anonymous Referee #2 · 19 May 2020

The authors should study the sensitivity of the Cloud Liquid Water Path (C-LWP) retrieval performance with respect to the cloud microphysics scheme used in simulation of the radar observations. Specifically, the simulated cloud water and rain variables are not completely independent and implicit statistical relationships (embedded in the cloud-observation simulation database) are exploited by the retrieval procedure to "separate" cloud water from rain. While the simulations used in the manuscript are realistic, it is conceivable that these implicit statistical relationships are not exactly the same as in nature. The additional study of different microphysics scheme would enable cross validation experiments and facilitate insight on the impact of the joint cloud water-rain distributions on the retrievals of the C-LWP.

---

## Referee Comment (RC2) · Matthew Lebsock (Referee) · 28 Jun 2020

GENERAL COMMENTS:

This paper is a timely study of the importance of integral constraints (Brightness temperature and path integrated attenuation) on the retrieval of cloud and rain liquid water path in warm rain. This paper is highly relevant to the NASA A/CCP mission preformulation activities which are ongoing and considering the effect of PIA/Tb on retrieval capabilities. Important insights into the effects of non-uniform-beam-filling in particular are shown. In particular the capability of along track oversampling to remove bias from the integral constraints is discussed.

The analysis method is appropriate. A couple of citations need to be added and some minor caveats need to be pointed out as described below.

SPECIFIC COMMENTS:

Lebsock and Suzuki, 2016 looked in detail at the ability of W-band PIA and Tb to constrain total water path in warm precipitation in a very similar study you should cite. https://doi.org/10.1175/JTECH-D-16-0023.1

page 8, line 10: Nothing is guaranteed. I would change the 'will' to 'must' or 'should'

page 9, line 5: 'Interestingly, TBs seem to react quicker than the PIA to the presence of the rain cells'. It is not really clear what this means. Do you mean quicker in terms of the Tb signal is larger than the PIA signal? In this case the quantity of interest is the sensitivity divided by the noise in each observation. Or do you mean responding quicker spatially along track? In this case be aware that the CloudSat Tb (as reported) has been averaged using a 5 pixel boxcar window so it has a resolution different than that of the PIA.

Page 10, line 16: The Marshall-Palmer isn't the best assumption for RICO rain. It will really overestimate drop size for a given rain water content. See for example Snodgrass (2009). This will cause you to overestimate the PIA and Tb enhancement so the signal will appear larger than it is likely to be in reality. At the least this caveat needs to be stated.

Page 11, line 12: Why add a constant 1 dB random noise to the PIA but use an SNR dependent noise for the reflectivities? The NRCS should be converted to a reflectivity and use the same SNR based noise that the profile uses. Or justify where this 1 dB number comes from.

Figure 7: colorbars have no units.

Page 12, line 25: 'defined as the antenna-weighted PIA'. I don't think this is stated correctly. I believe you are referencing the PIA of the precipitating or cloudy part of the

antenna pattern while neglecting clear sky. When I read 'antenna-weighted' that seems to include the clear sky. If it did include the clear sky the SRT would not underestimated the PIA. Defining quantities through equations could eliminate the ambiguity.

Page 13, line 15: The conclusion here regarding minimum LWP sensitivity is a function of your assumption of 2K Tb uncertainty. I think you ought to state that explicitly as there is no reason a radar with better Tb precision cannot be designed (which in practice would likely mean sacrificing minimum reflectivity sensitivity). For example Lebsock and Suzuki (2016) show for this RICO case that 10 gmˆ-2 TWP sensitivity should be achievable with different Tb uncertainty assumptions.

-Reviewed by Matt Lebsock-

REFERENCES:

Lebsock M.D. and Kentaroh Suzuki, 2016: Uncertainty Characteristics of Total Water Path Retrievals in Shallow Cumulus Derived from Spaceborne Radar/Radiometer Integral Constraints. J. Atmos. Oceanic Technol. 33, 1597–1609, doi: 10.1175/JTECH-D-16-0023.1.

Snodgrass, E. R., L. Di Girolamo, and R. M. Rauber (2009), Precipitation characteristics of trade wind clouds during RICO derived from radar, satellite, and aircraft measurements, J. Appl. Meteorol. Climatol., 48, 464–483, doi:10.1175/2008JAMC1946.1.

---

## Author Comment (AC1) · 25 Jul 2020

The authors would like to thank very much the reviewer for his useful and insightful feedbacks. A point-by-point response to the reviewer's comments is provided below. Refer to the bits in bold face in the paper manuscript attached at the end of this document to see how the comments have been addressed.

**Reviewer 1**

**Specific comments:**

**Lebsock and Suzuki, 2016 looked in detail at the ability of W-band PIA and Tb to constrain total water path in warm precipitation in a very similar study you should cite. https://doi.org/10.1175/JTECH-D-16-0023.1**

Yes absolutely! we were not aware of the paper, which provides some key analysis of the potential of a radiometric mode with CloudSat measurements and some very useful considerations. We have reshaped the paper including reference and discussion to the results of such paper.

**Page 8, line 10: Nothing is guaranteed. I would change the 'will' to 'must' or 'should'**

Yes agreed and indeed we may see a worsening in some of the radar performances (e.g. footprints) in the next generation of sensors (this for instance seems the trend from the NASA ACCP studies).

**page 9, line 5: 'Interestingly, TBs seem to react quicker than the PIA to the presence of the rain cells'. It is not really clear what this means. Do you mean quicker in terms of the Tb signal is larger than the PIA signal? In this case the quantity of interest is the sensitivity divided by the noise in each observation. Or do you mean responding quicker spatially along track? In this case be aware that the CloudSat Tb (as reported) has been averaged using a 5 pixel boxcar window so it has a resolution different than that of the PIA.**

Yes agreed. Here we were pinpointing at the greater sensitivity of TBs. A sentence to better explain what we meant has been now introduced. A section on sensitivity has now been introduced later in the paper (Sect3.2.2).

**Page 10, line 16: The Marshall-Palmer isn't the best assumption for RICO rain. It will really overestimate drop size for a given rain water content. See for example Snodgrass (2009). This will cause you to overestimate the PIA and Tb enhancement so the signal will appear larger than it is likely to be in reality. At the least this caveat needs to be stated.**

The reviewer's point is important. Rain microphysics is indeed relevant. In the SAM model actually a two-moment scheme is adopted where rain concentration and rain water contents are both predicted (Morrison et al, 2005). All other moments of the DSD can be derived by using the assumption that DSDs are exponential distributed. An exponential fit of the form $Lambda^{-1} = 2.7 * 10^{-4} * RWC^{0.3}$ (like proposed in Lebsock and L'Ecuyer, 2011) where Lambda is the slope parameter in $m^{-1}$ and RWC is the water content in $kg/m^3$ is used in our case (and it will be referred as MO05). Alternatively we have used two other configurations, one with $N_0 = 8 * 10^6$ $m^{-4}$ (Marshall and Palmer) and the other with $N_0 = 5 * 10^7$ $m^{-4}$ which resembles the Cumulus parametrization used in Lebsock and L'Ecuyer, 2011. See new Fig.11 and new text in Sect. 3.2.2.

Of course information about rain microphysics could be grasped from a full profile multi-frequency retrieval (properly constrained by integral constraints in order to properly account for attenuation). For instance the effective (i.e. attenuation corrected) DFR Ka-W or W-G band have very good sensitivity to the rain microphysics as demonstrated by Fig.A3 right panel in Battaglia et al., 2020.

**Page 11, line 12: Why add a constant 1 dB random noise to the PIA but use an SNR dependent noise for the reflectivities? The NRCS should be converted to a reflectivity and use the same SNR based noise that the profile uses. Or justify where this 1 dB number comes from.**

Yes this is correct when considering only the noise of the surface reflectivity measurements (which is 0.16 dB at high SNR). Indeed as shown in Eq.21 of Leinonen et al, 2017 the uncertainty in the PIA is dominated by the uncertainty the clear sky sigma0 not by the uncertainty of the surface return (unless for cases closes to full attenuation). Leinonen et al., 2017 claim that the former uncertainty can be estimated from the standard deviation of the observed clear-sky surface cross sections, with an average value of 0.24 dB so that the total PIA precision is estimated to be 0.29 dB. In reality this is probably a very optimistic assumption because the clear sky sigma0 is likely more uncertain, especially if there are no close-by ``clear sky'' observations (especially when continuous Sc decks are present). In addition, winds are likely to change in presence of precipitation due to local circulation (see Figure 1) and the sigma0 has a strong sensitivity to the 10-m wind speed (whereas the sensitivity to SST is weaker) like shown in Figure 2.

[Figure]

**Figure 1: hydrometeor content and 10-m wind speed as simulated by the SAM model for one cloud scene. Clearly, the wind field shows quite some patchiness with some strong gradients corresponding to the cloud/rain edges.**

E.g., with 7-10 m/s wind (which is a characteristic value in our simulations), an uncertainty of 1m/s in the wind speed causes an uncertainty of roughly 0.6-0.4dB in the sigma0 (with much worse

results obtained for lower wind speeds). Therefore we generally think that a PIA precision not smaller than 0.7 dB is more appropriate to be assumed in our discussion. Text in Sect.3.1.2 has been modified accordingly.

[Figure]

**Figure 2: sensitivity of sigma0 to the 10-m wind speed (left) and the SST (right). The model by Wu et al., has been used.**

**Figure 7: colorbars have no units.**

The caption to the figure has now been updated to address that.

**Page 12, line 25: 'defined as the antenna-weighted PIA'. I don't think this is stated correctly. I believe you are referencing the PIA of the precipitating or cloudy part of the antenna pattern while neglecting clear sky. When I read 'antenna-weighted' that seems to include the clear sky. If it did include the clear sky the SRT would not underestimated the PIA. Defining quantities through equations could eliminate the ambiguity.**

Quantities have been defined in the paper and hopefully things are clear now.

**Page 13, line 15: The conclusion here regarding minimum LWP sensitivity is a function of your assumption of 2K Tb uncertainty. I think you ought to state that explicitly as there is no reason a radar with better Tb precision cannot be designed (which in practice would likely mean sacrificing minimum reflectivity sensitivity). For example Lebsock and Suzuki (2016) show for this RICO case that 10 gm^-2 TWP sensitivity should be achievable with different Tb uncertainty assumptions.**

Agreed and this has been now discussed in the conclusions. See new text in Sect.4.

[revised manuscript text omitted]

---

## Author Comment (AC2) · 25 Jul 2020

The authors would like to thank very much the reviewer for his useful feedback. The paper has been modified accordingly (see link in Reply to Reviewer 1). Reviewer 2 Specific comments: The authors should study the sensitivity of the Cloud Liquid Water Path (C-LWP) retrieval performance with respect to the cloud microphysics scheme used in simulation of the radar observations. Specifically, the simulated cloud water and rain variables are not completely independent and implicit statistical relationships (embedded in the cloud-observation simulation database) are exploited by the retrieval procedure to "separate" cloud water from rain. While the simulations used in

the manuscript are realistic, it is conceivable that these implicit statistical relationships are not exactly the same as in nature. The additional study of different microphysics scheme would enable cross validation experiments and facilitate insight on the impact of the joint cloud water-rain distributions on the retrievals of the C-LWP. We agree with the reviewer that the implicit relationship between C-LWP and R-LWP in the training database can indeed introduce additional uncertainties to the retrieval. Following a similar remarks from Rev1 on rain microphysics we have now introduced a sensitivity analysis in Sect.3.2.2 to show the effect of selecting different DSD parametrizations. A new figure (11) has been introduced. Studying the effect of different combinations of rain/cloud partitioning seems more appropriate in a full retrieval study where the integral constraints should be used in combination with the full reflectivity information and is beyond the scope of this work. This has been noted in the conclusions.